# Circulating Tumor Cells in Colorectal Cancer: Detection Systems and Clinical Utility

**DOI:** 10.3390/ijms232113582

**Published:** 2022-11-05

**Authors:** József Petrik, Donatella Verbanac, Marija Fabijanec, Andrea Hulina-Tomašković, Andrea Čeri, Anita Somborac-Bačura, Roberta Petlevski, Marija Grdić Rajković, Lada Rumora, Božo Krušlin, Mario Štefanović, Neven Ljubičić, Neven Baršić, Antonija Hanžek, Luka Bočkor, Ivana Ćelap, Alma Demirović, Karmela Barišić

**Affiliations:** 1Department of Medical Biochemistry and Hematology, Faculty of Pharmacy and Biochemistry, University of Zagreb, Ante Kovačića 1, 10000 Zagreb, Croatia; 2School of Medicine, University of Zagreb, Šalata 3, 10000 Zagreb, Croatia; 3Department of Pathology and Cytology “Ljudevit Jurak”, University Hospital Centre “Sestre milosrdnice”, University of Zagreb, Vinogradska 29, 10000 Zagreb, Croatia; 4Department of Clinical Chemistry, University Hospital Centre “Sestre milosrdnice”, University of Zagreb, Vinogradska 29, 10000 Zagreb, Croatia; 5Department of Internal Medicine, University Hospital Centre “Sestre milosrdnice”, Division of Gastroenterology and Hepatology, University of Zagreb, Vinogradska 29, 10000 Zagreb, Croatia; 6School of Dental Medicine, University of Zagreb, Gundulićeva 5, 10000 Zagreb, Croatia; 7UPR CHROME, University of Nimes, 7 Place Gabriel Peri, 30000 Nîmes, France; 8Centre for Applied Bioanthropology, Institute for Anthropological Research, Ljudevita Gaja 32, 10000 Zagreb, Croatia

**Keywords:** colorectal cancer, liquid biopsy, circulating tumor cells, CTC clusters, viable CTCs, isolation techniques, detection methods, clinical utility

## Abstract

Colorectal cancer (CRC) is the third most common cancer worldwide. The high mortality from CRC is mainly related to metastasis affecting distant organs and their function. Dissemination of tumor cells from the primary tumor and hematogeneous spread are considered crucial in the formation of tumor metastases. The analysis of circulating tumor cells (CTCs) and CTC clusters in the blood can be used for the early detection of invasive cancer. Moreover, CTCs have a prognostic significance in the monitoring of a malignant disease or the response to chemotherapy. This work presents an overview of the research conducted on CTCs with the aim of finding suitable detection systems and assessing the possibility of clinical applications in patients with CRC.

## 1. Introduction

Colorectal cancer (CRC) is the third most common cancer in the world, after breast and lung cancer. According to the estimation by the World Cancer Research Organization, almost two million people are diagnosed annually, from which 0.95 million die. The prevalence of the disease is increasing due to the large number of patients diagnosed and the increasingly better survival rates in the last decade [1].

According to the published data of the Croatian Institute of Public Health from 11 March 2022 [2], colon and rectal cancer is the most common malignant disease in Croatia. On average, about 3600 patients are diagnosed per year, of which about 60% are men. It is more common in older adults; however, almost a fifth of the patients are younger than 60 years old. According to the latest data from the Cancer Registry in 2019, 3660 people were diagnosed with colorectal cancer, with the average age at the time of diagnosis being 69. In terms of mortality from malignant conditions, colorectal cancer is second only after lung cancer. About 2100 people die from it annually, 60% of them being men. The mortality trend has been stable in the last ten years, but the incidence of CRC has been increasing by about 1% per year over the previous 20 years. According to the estimates of the European Commission, CRC incidence rates are highest in Slovakia, Denmark, Hungary, and the Netherlands, whereas mortality is the highest in Eastern European countries. Croatia is in ninth place in terms of colon cancer incidence and in second place in terms of mortality. These data indicate that there is much room for improvement at all levels of colorectal cancer prevention. If colon cancer is detected in a localized stage, the five-year survival rate is about 90%. However, according to the latest data from the Cancer Registry, only 11% of colon cancer cases were detected in a localized stage. In comparison, distant metastases were already present in 13% of patients at the time of diagnosis. In the international study on survival (CONCORD-3), the five-year survival rate for patients diagnosed with colon cancer between 2010 and 2014 in Croatia was 51.5%, and 48.2% for patients diagnosed with rectal cancer, which indicates that Croatia is falling behind countries in northern and western Europe, where the survival rate is around 65% [2,3].

Of all malignant diseases, approximately one-third of deaths are related to gastrointestinal (GI) cancers, such as cancers of the oesophagus, stomach, pancreas, colon, and rectum [4].

The formation of the metastases leads to severe complications and endangers the patient’s life. Clinical or biochemical suspicion of metastatic disease should always be confirmed by appropriate radiological imaging (usually computed tomography (CT), magnetic resonance imaging (MRI) or ultrasound). Fluorodeoxyglucose-positron emission tomography (FDG-PET) scanning can be helpful in determining the malignant characteristics of tumor lesions, especially in combination with CT imaging or in the case of elevated tumor markers (carcinoembryonic antigen (CEA)) without indications of a relapse site on CT imaging in CRC supervision. FDG-PET imaging is also particularly useful for characterizing the extent of metastatic disease and looking for extrahepatic metastases (or extrapulmonary metastases) when the metastases are potentially resectable [5]. Immunohistochemistry (IHC) is used to identify CRC primary tumors of unknown origin. CRCs express the nuclear transcription factor CDX2, which is highly specific for intestinal epithelial cells, and villin, which is typical for adenocarcinomas of the GI tract. Additionally, *GPA33* encodes a membrane protein expressed in most colorectal tumors and is an effective marker, especially for well-differentiated cancers, whereas it is also a recent target for antibody radioimmunotherapy [6,7]. Other biomolecules which are helpful in identifying the origin of colon adenocarcinoma include proteins such as CEA, calretinin, CDH17, MOC-31 (Ep-CAM), CK20, CK7, and, CA19-9, as well as enzymes such as MUC2. Due to mutations in the *KRAS, BRAF,* and *PIK3CA* genes, a significant ratio of colorectal cancers also have altered protein products, which could be used for IHC diagnosis [8].

The appearance of micrometastases is associated with an increase in the number of circulating tumor cells (CTCs) in the blood of patients with cancer, carrying with them a plethora of antigens which can be used for diagnostic purposes [9,10]. The role of CTCs in tumor metastasis is shown in Figure 1.

Recent research papers [11,12,13] highlight the need for additional procedures and biomarkers in the field of precision medicine. Future steps in the clinical treatment of metastatic colorectal cancer (mCRC) will include the integration of the comprehensive knowledge of tumor gene alterations, gene expression profiling of the tumor and microenvironment, proteomic research, and host immune capabilities. These applications will result in a dynamic and persistent change in precision medicine [14].

Carcinomas are very heterogeneous tissues composed of histologically different cells. In addition to neoplastic cells in or around the tumor, there are also nonmalignant cells forming a tumor-altered microenvironment. Tumor growth and its invasiveness depend on the malignant transformation of the cells that form the microenvironment and the patient’s immune response [15].

Using transcriptome analysis, Shaath and colleagues identified several differentially expressed transcripts that reflect the systemic immune cell effect on tumorigenesis, which have the potential use as disease biomarkers. Interestingly, peripheral blood mononuclear cells (PBMCs) from CRC patients showed an induction of the inflammation and immune response. The potential use of changes in the circulating immune cells as indicators of CRC requires further investigation [16].

CRC mortality is often associated with metastases affecting other organs’ functions. The hematogenous spread of tumor cells is considered crucial in tumor metastasis [9,14,17]. Therefore, the cancer treatment primarily focuses on the prevention of metastasis formation and treatment of the resulting metastatic processes in the early phase [18].

Since radiological and endoscopic imaging techniques have a low level of sensitivity for detecting the micrometastatic process, other diagnostic tools, such as liquid biopsy, are being seriously considered and implemented in the current practice. Liquid biopsies provide the possibility of tumor analysis, which includes protein and DNA/RNA analysis, i.e., molecular profiling after CTC isolation [12,19]. Examples include disseminated tumor cells (DTCs) in the bone marrow and CTCs in the peripheral blood, which are increasingly used as prognostic markers. Several studies have investigated their effectiveness in diagnosis [20,21].

The lack of a satisfactory tool for the early diagnosis of colorectal cancer is the main reason for the late diagnosis, often at an advanced stage of the disease. Furthermore, there is a lack of reliable biomarkers that could predict clinical features, such as cancer invasiveness. Consequently, the search for tumor markers that can be reliably used, taking into account preanalytical, analytical, and postanalytical requirements; the level of evidence; and the strength of recommendation in certain stages of colorectal cancer, remain the focus of research.

This paper presents research on CTCs to find suitable detection systems and the possibility of their clinical application in patients with CRC.

## 2. Liquid Biopsy

Liquid biopsy is a minimally invasive technique for detecting molecular biomarkers that uses an analysis of liquid biological material (blood, urine, ascites, etc.) [22]. Back in 1948, Mandel and Métais [23] first described the presence of circulating free nucleic acids (cfNAs) in the human blood. Many solid tumors and metastases release biomarkers into systemic circulation, such as signaling molecules, macromolecules, nucleic acids, exosomes, and fragments of cells, or even CTCs [22,24,25,26].

The cfDNAs are discharged into the circulation from damaged cells by various mechanisms, but primarily by apoptosis and necrosis. In cancer patients, the fraction of cfDNA originating from the tumor is called ctDNA, consisting of DNA fragments between 120 and 200 bp in size. The primary challenge of ctDNA analysis is the low concentration of ctDNA compared to the total DNA present in serum. Depending on the tumor stage and the response to therapy, the concentration of the ctDNA fraction can vary from 0.01% to 90% of the total cfDNA [19]. Isolated ctDNA from plasma carries genetic and epigenetic changes originating from the primary tumor and enables the molecular analysis of mutations. Clinically, it is used as a biomarker for patient stratification, therapy selection, and monitoring of the effectiveness of the therapy, especially in lung and colorectal cancer patients [27,28]. Monitoring of the patients using ctDNA can detect resistance to treatment in real-time, which was shown in patients with breast, ovarian, and lung cancer [29].

Exosomes are subcellular membrane-enveloped vesicles with a diameter of 30–150 nm, consisting of proteins, microRNA (miRNA), and DNA, which can provide information about the primary tumor and the microenvironment. They are identified in high concentrations in blood, ascites, and cerebrospinal fluid, and even in tumors that do not have measurable levels of CTCs in circulation. Exosomes are stable, and thus, they can be analyzed using stored or frozen samples. Research has identified new exosomal miRNA markers in lung, prostate, and stomach cancer and demonstrated the importance of exosomal DNA for tumor molecular analysis [30]. miRNAs are noncoding RNA molecules that function as regulatory molecules of gene expression. Although they are distributed through the whole genome, most of them are located in fragile genomic regions that are prone to deletions and are related to different cancers. Therefore, miRNA alternations can be associated with tumorigenesis and cancer progression. The most commonly identified miRNA biomarkers are in colon, lung, breast, and skin cancer. However, the current review of the literature does not suggest miRNA as a diagnostic tool for liquid biopsy in clinical practice, primarily due to the lack of adequate technology and standardization [31].

The main advantage of liquid biopsy is the availability of material for diagnosis. Blood and urine are routinely extracted in biomedical labs, and more complex fluids also have standardized, readily available extraction techniques.

Tumor tissue biopsies are invasive, painful, time-consuming, and carry a specific risk of complications. They are not feasible at progressive stages of the disease or when the localization of the primary tumor is not known. Liquid biopsy is always applicable precisely because of the simplicity of peripheral blood sampling and enables less invasive detection, characterization, and monitoring of tumors [32].

Identifying specific mutations in target genes is essential in clinical decision-making when characterizing the tumor and choosing the right therapy. The heterogeneity of somatic tumor mutations is a challenge for treatment due to the possibility that a single tissue biopsy may not reveal a relevant lesion for the application of the appropriate therapy. Therefore, the temporal and spatial limitations of tissue biopsy are evident in the emergence of acquired resistance to therapy. Liquid biopsy provides better insight into the genome of the entire primary tumor and metastases [19]. One more limitation of conventional tissue biopsy is a lack of longitudinal disease monitoring and treatment response. The unstable tumor genome is known to change dynamically over time in response to treatment, resulting in the suppression or enhanced growth of specific cell clones. Liquid biopsy enables real-time tumor analysis [26]. It is, therefore, suitable for the serial monitoring of patients at any time point in the course of the disease or remission, allowing for less invasive insight into tumor heterogeneity [32].

## 3. Circulating Tumor Cells

CTCs are intact cells separated from the primary tumor or metastases and released into the peripheral circulation. They were observed and discovered for the first time in 1869 in the blood of a patient with breast cancer [33]. CTCs mainly originate from solid tumors of epithelial origin (breast, prostate, colon, and lung). CTCs are nucleated and express epithelial cell adhesion molecules (EpCAM) and/or cytokeratins (CK) in the cytoplasm without coexpressing the common leukocyte antigen CD45. It is known today that there is significant heterogeneity in cell species and surface markers, which represents a challenge in isolating all clinically relevant subpopulations of CTCs. The presence of CTCs is associated with a worse prognosis and a high probability of the occurrence of a metastatic process. However, adequate detection and characterization of CTCs can significantly contribute not only to the diagnosis of cancer, but also to the prediction of the outcome of the disease, the selection of the optimal therapy, and the monitoring of the therapeutic treatment (Figure 2).

The analysis of CTCs from blood can be used for the early detection of invasive cancer. Moreover, they have a prognostic significance in the monitoring of malignant disease or the response to chemotherapy. CTCs are sporadic cells with a frequency of one tumor cell per 5 × 106 leukocytes and 5 × 109 erythrocytes per mL of blood in patients with advanced cancer [34]. Numerous isolation technologies have been developed that enable the isolation and differentiation of CTCs from normal blood cells. Compared to blood cells, CTCs are more rigid, with a density of >1.077 g/mL, and larger in diameter than leukocytes, which have an average range of 8–11 μm. CTCs are relatively large cells, with an example of 30 μm in size in breast cancer patients [35].

With the degradation of the tumor mass in the first stages of the metastatic cascade, the tumor cells are released and migrate to the distal parts of the body by lymphohematogenous dissemination. During circulation and colonization, various interactions with other cells can lead to changes in CTC phenotypes [36]. The majority of CTCs die, but 0.01% stay viable and have the potential to form metastases. They may be present as single cells or clusters of cells. Once localized in organs (bone marrow, liver, lung, or brain), they are called disseminated tumor cells (DTCs) [35]. DTCs and micrometastases can be in a dormant phase for years after the primary tumor resection. At some point, DTCs can proliferate under the influence of the tumor microenvironment, and the formation of metastases occurs. DTCs originating from metastases can recirculate through the blood, colonizing other organs and creating secondary metastases. Research suggests that DTCs can transform into CTCs and return to the primary tumor site, resulting in the formation of more aggressive variants of tumor metastases [22].

Noninvasive liquid biopsy opens up detection possibilities for very rare circulating cancer stem cells (CSCs). CSCs and CTCs are particularly interesting targets in basic research, but also in clinical studies, due to their multiple roles and involvement in tumorigenesis, cancer progression, and resistance to therapy [37,38,39]. However, it is currently difficult to use these cells in the clinical setting: firstly, due to the lack of fully validated biomarkers, and secondly, due to the nonunique method for their identification. In fact, there is still no specific single marker or combination of markers that represent these cell subpopulations and clearly distinguishes them from conventional cancer cells. Indeed, CSCs are known to constitute only a small percentage (0.05–1%) [40,41] of tumor cells. The difficulty of CTC analysis is cellular plasticity, especially the phenomenon of the epithelial–mesenchymal transition (EMT). The process of the epithelial–mesenchymal transition is the change from the epithelial to mesenchymal phenotype. The EMT is a complex process of cell dedifferentiation and increased cell mobility due to the reorganization of contact bonds and the loss of adhesion molecules. Although this mechanism primarily occurs in organogenesis and wound healing, it is associated with tumor dissemination and correlates with aggressiveness due to increased tumor cell migration. The opposite process is also possible (mesenchymal–epithelial transition), which is proposed to play a role in the transformation of dormant DTCs in organs to metastases. This can lead to the nondetection of the population of potentially aggressive tumor cells in the standard CTC diagnostic process, as it relies on epithelial markers [22]. New technologies that could separate different CTC phenotypes are needed, highly relevant, and essential [42].

## 4. Diagnostic Significance of Circulating Tumor Cells

Developing new methods that enable the early diagnosis and follow-up for patients in all stages of the disease is crucial for reducing mortality and improving the outcome of CRC. The main objectives of research on CTCs as multifunctional biomarkers include: (1) identification of therapeutic targets, (2) stratification of oncological patients and monitoring of therapy in real-time (predictive significance), (3) a risk assessment for metastatic relapse or disease progression (prognostic significance), and (4) understanding the tumor biology and mechanisms of resistance to therapy [13,22].

Molecular characterization of CTCs provides a noninvasive approach to tumor genotypic and phenotypic characteristics. Treatment decisions are empirically defined by tumor histology after biopsy, which is sometimes performed after the initial treatment decision. Characterizing tumors from CTCs using a simple blood test could enable more effective treatment choices and reduce the incidence of unnecessary toxicity and side effects in patients receiving chemotherapy and immunotherapy [43]. Detection of mutations based on CTCs in the blood provides information that could significantly accelerate the discovery of a therapeutic target. Compared to tissue biopsy, a shorter turnaround time (TAT) of molecular analysis leads to faster therapeutic intervention and better disease outcomes. Since the tumor mutational profile affects the choice of therapy, research has been performed on colorectal cancer [44], lung cancer [45], breast cancer, and melanoma [46,47]. Table 1 shows the clinical significance of CTC determination in patients with CRC. The table also shows values such as the proportion of patients with positive CTC findings and the clinical usefulness of the determination [48,49,50,51,52,53,54,55,56,57,58,59,60,61,62,63,64,65],

The predictive significance of CTC analysis includes the detection of CTCs in the early stage of malignancy in resectable solid tumors. The goal is to monitor the effectiveness of the therapy and select patients who would benefit the most from adjuvant treatment. Furthermore, tracking the number of CTCs at the end of therapy in patients’ follow-ups could detect relapses earlier than radiological parameters. Furthermore, it is necessary to highlight the importance of CTC clusters for future diagnostic and therapeutic approaches. CTC clusters have great importance in the development of metastases, and therefore, their enumeration and characterization stands as a potential target for future clinical application [66].

## 5. Methods and Technologies of Isolation Circulating Tumor Cells

CTC isolation methods are based on the differences between target CTCs and blood cells. These differences include physical properties (size, density, electrical charge, and deformability) and specific biological characteristics (expression of surface protein markers and viability) [22,49,51,54,59,62].

CTC isolation and enrichment based on biophysical characteristics are gaining popularity due to antibody-independent isolation. The most common methods for analysis are using optical microscopy and/or flow cytometry based on the difference in morphological characteristics and size of CTCs compared to leukocytes. Centrifugation is one of the first methods for CTC isolation. The OncoQuick^®^ procedure combines density gradient centrifugation with microporous membrane filtration for cell separation. Although a simple and inexpensive method, it is used as the initial step of the CTC analysis, often in combination with other systems, due to leukocyte contamination.

The most commonly used isolation principle is immunoaffinity. Immunoaffinity methods include the detection of specific markers (antigens) on the cell surface. The prognostic significance of CTC analysis has been confirmed in different types of tumors. The number of CTCs in the blood is an independent predictor of overall survival (OS) and progression-free survival (PFS) in patients with metastatic breast, prostate, and colorectal cancer [47]. This area of research is the most active and promising, thanks, in part, to the development of the CellSearch^®^ (Janssen Diagnostics, Veridex, South Raritan, NJ, USA) technology. CellSearch^®^ is the first clinically validated and approved liquid biopsy technology for the enumeration of CTC analysis from peripheral blood, providing valuable information for making clinical decisions at any time during patient monitoring. It includes sample collection, preparation, and analysis using the immunomagnetic analysis principle and fluorescence imaging technology [47]. The method was approved by the Food and Drug Administration (FDA) in 2004 for monitoring patients with metastatic breast cancer, in 2007 for monitoring patients with colorectal cancer, and in 2008 for monitoring metastatic prostate cancer.

The CellSearch^®^ system consists of a semiautomated device and corresponding components. The sample for CTC analysis is whole blood at a volume of 7.5 mL. Because CTCs are sensitive and are subject to degradation, blood is collected in specialized CellSave tubes (Jannsen Diagnostics, Raritan, NJ, USA), which allow for the stabilization of target cells at room temperature. Samples are processed up to 96 h after collection in classic CellSave tubes. In the case of viable cell analysis or studies of RNA expression in CTCs, blood sampling using EDTA and processing within 24 h is recommended [32]. The principle of the method is immunomagnetic analysis. The technology uses ferrofluidic nanoparticles functionalized with EpCAM antibodies that enable magnetic separation of EpCAM-positive cells from other blood components after centrifugation. Then, immunofluorescence labeling of CTCs is performed using antibodies against cytokeratins (CK 8, 18, and 19), which are specific for tumor epithelial cells, and antibodies against CD45, a common leukocyte antigen. Finally, cell nuclei are stained with DAPI (4’,2’-diamidino-2-phenylindole dihydrochloride). Samples are prepared for analysis using a magnetic separation cassette. The magnetic field allows cells to be separated so that each cell passes individually through a fluorescence signal detector. The system selects EpCAM+/CK+/CD45-/DAPI+ cells as potential CTC candidates. The candidates are presented through a software platform, which requires an experienced user for the interpretation of the results. The results are recorded as the number of CTCs/7.5 mL of whole blood [47,67]. This technology preserves the viability and proliferative and invasive potential of CTCs. Cells of interest are isolated alive and can be maintained in culture, allowing for expansion and ex vivo characterization of CTC subpopulations. This enables capturing CTCs with metastatic potential, as only viable cells can rise to metastasis. Following these isolation technologies, a functional test called the Epithelial ImmunoSPOT test (EPISPOT) can be used that selects viable CTCs based on the detection of specific secreted tumour-associated proteins [68]. The EPISPOT assay enumerates only viable CTCs, regardless of EpCAM expression, because this innovative technology is always combined with leukocyte depletion [69]. Furthermore, in nonmetastatic CRCs, the CK19-EPISPOT assay detected more CTCs than the CellSearch^®^ system in the peripheral and mesenteric blood samples from patients with previously untreated tumors [70]. The EPISPOT test can also be used to successfully monitor the clinical importance of vital circulating tumor cells in patients with metastatic colorectal cancer [68].

The AdnaTest (AdnaGen AG) is a similar CTC detection platform, where isolation and enrichment are achieved using magnetic particles coated with antibodies. However, the AdnaTest uses multiple specific antibodies for a particular type of tumor (breast, prostate, ovarian, or colon cancer) [42].

The MagSweeper (Stanford University, The Office of Technology Licensing, Stanford, CA, USA), an immunomagnetic isolation and enrichment technology, can isolate high-purity CTCs. The functional part of the device is a robotically controlled magnetic arm that isolates CTCs from the sample. The sample is mixed with magnetic particles coated with tumor antibodies. The magnetic rod (arm) passes in circular motions over the samples and isolates the magnetically labeled CTCs. The isolation procedure consists of capturing the desired labeled tumor cells, whereas other contaminating cells are released and washed. The process is repeated until high-purity CTCs are obtained [71].

CTCs can also be analyzed using microchips and microfluidic devices, such as the CTC Chip, GEDI, and OncoCEE. Microfluidic devices, the analytical “lab on a chip” systems, have integrated all the necessary components for sample analysis. The first CTC-Chip microfluidic device designed for CTC isolation was developed in 2007. It consists of a microarray that is chemically functionalized with anti-EpCAM antibodies. The researchers optimized the geometric arrangement of the device surface and the fluid flow control for efficient CTC capture on the device surface. With this technology, CTCs were successfully isolated from the peripheral blood of patients with metastatic disease (lung, breast, prostate, colon, and pancreas) [42].

GEDI immunochemical microfluidic technology was tested on patients with resistant prostate cancer. The system is a combination of positive selections using a microarray with antibodies and specific prostate membrane antigens and hydrodynamic chromatography, which isolates cells based on size. The study was conducted with the aim of molecular characterization of the isolated CTCs to discover new targets for antitumor therapy [72].

In addition to the low concentration of CTCs in blood, the challenge for CTC analysis can be low or insufficient blood volume in oncology patients. Therefore, devices for direct in vivo isolation of CTCs have been developed, such as the GILUPI nanodetector. During the 30 min application of the device in the forearm vein, a large volume of blood passes through the device. Inside the device, up to 1.5 L of blood can pass through the “screening” area coated with EpCAM antibodies, resulting in the successful binding of the CTCs. The surface of the nanodetector can be coated with specific tumor antibodies, and CTCs are removed for downstream immunocytochemical or molecular analysis. A study with clinical samples from breast and lung cancer patients showed that the GILUPI device succeeds at isolating CTCs. The device is biocompatible and does not have known side effects after application [73].

## 6. Strategies for Characterization of Circulating Tumor Cells

After CTC isolation, various CTC detection strategies are applied to eliminate leukocytes and confirm at the cellular level that they are indeed CTCs. There are two basic approaches for the detection and characterization of CTCs; protein-based methods and nucleic acid-based methods. Immunofluorescence and flow cytometry are protein-based methods of CTC detection and characterization. The most common method for protein-based detection of CTCs is antibody-based immunofluorescence. The advantages of immunofluorescence are the visual confirmation of protein expression and its localization, the possibility of a simultaneous analysis of multiple proteins, and the quantification of protein expression. The disadvantages are a low sensitivity and low commercial availability of specific antibodies for CTCs. In flow cytometry, fluorescently labeled CTCs pass through a beam of laser light. Information about an individual cell is obtained based on light scattering and fluorescence, describing the cell’s shape, size, granularity, and marker expression, which enables cell classification. The advantages of flow cytometry are multimarker analysis at the single-cell level, the possibility of sorting CTCs into subpopulations, and the availability of the method in most routine hemato-oncology labs. On the other hand, the main disadvantages are a low sensitivity for rare CTC populations and the inability to distinguish between CTCs and leukocytes [74].

Methods based on nucleic acids are standard in the field of molecular biology, namely: RT-PCR, fluorescent in situ hybridization (FISH), and sequencing. These methods aim to detect specific tumor transcripts, which can confirm the presence of CTCs in the background of other blood cells. RT-PCR is used to detect target transcripts and characterize CTCs (e.g., for the determination of HER2, ER, and PR status in breast cancer). The advantage of these methods is the possibility of analyzing several genes simultaneously from a small sample volume, which is necessary for CTC research. FISH can be used to detect and characterize CTCs to detect changes in individual genes or chromosomes [54]. Sequencing is considered an ideal choice for CTC characterization because it can use genomic DNA and cfDNA and analyze numerous target genes simultaneously. Therefore, it is a powerful tool for analyzing specific genomic aberrations. An advantage of CTC genome sequencing is the ability to detect changes at the level of a single nucleotide, as such mutations can result in different disease phenotypes and can alter the response to therapy. The disadvantage of sequencing is the fact that at least 50 CTCs are required for a valid result, meaning that it is impossible to locate a sequence back to a specific sample and visually confirm the finding [74].

## 7. Assessment of ScreenCell Cyto Kit

In this paper, we present the partial results obtained by testing the blood of colorectal cancer patients. The patients were included in the research within the project financed by the Croatian Science Foundation, grant number IP-2019-04-4624 (project “CRCMolProfil—Genetic, protein and RNA profiling of colorectal cancer using liquid biopsy”). The patients signed the informed consent form, and we performed the tests according to the standards of the Good Clinical Practice, Good Laboratory Practice, and the Helsinki Declaration on Ethical Principles, which were considered in the study design. We used the ScreenCell^®^ (Sarcelles, France) Cyto kit system for the isolation of CTCs from the patient’s whole blood, following the recommendations of the manufacturer ScreenCell^®^ SA, France (CYTO Réf. CY 4FC, 2017). This system is designed to detect and determine the number of CTCs. After filtration of the diluted blood, CTCs and CTC clusters are found on the porous membrane of the ScreenCell^®^ Cyto system if they are present in the blood sample. Additionally, some larger atypical cells, microcoagulum, and a lesser number of leukocytes can also be found. After filtration, the preparations were fixed and analyzed. Cells were analyzed based on the morphological characteristics of cells by May–Gruenwald–Giemsa staining (Biognost—BIO-DIFF RTU KIT) or based on the presence of cytokeratin on cells of an epithelial origin using a primary antibody (monoclonal mouse anti-human cytokeratin clone AE1/AE3 (Dako, IR053)) and secondary antibody (dextran coupled with peroxidase molecules and goat secondary antibody molecules against rabbit and mouse immunoglobulins in a buffered solution containing stabilizing protein and preservative (Dako, K8000)). CTCs were also detected using the immunofluorescence method by determining the cytokeratin, CD45 antigen, and DAPI according to the methods described by [62]. Double immunofluorescence staining was performed with a primary mouse anti-pan-CK (AE1/AE3, #M3515; Agilent Technologies Inc., Santa Clara, CA, USA), anti-CTC antibody, and rabbit anti-CD45 (EP68) antibody (#AC-0065A; Epitomics Inc., Cambridge, UK). In the final stage, secondary antibodies were used for visualization: goat-anti-mouse and Alexa488 (green—against CTC) (#A11001; Life Technologies, Carlsbad, CA, USA) and goat-anti-rabbit Alexa568 conjugates (red—against leukocytes) (#A110; Life Technologies Corporation). CTCs were positive for DAPI and Alexa488, and negative for Alexa568. The obtained results are shown in Figure 3 and Figure 4. In this study, the Olympus BX50 fluorescence microscope with DPSoft (Olympus Optical Co. Ltd., Tokyo, Japan) and the ImageXpress Micro Confocal System (Molecular Devices, LLC, San Jose, CA, USA) were used.

## 8. Discussion

The low concentration of CTCs in the blood and the great diversity of CTC subpopulations make the development of a new diagnostic method very demanding (Figure 5). CTCs are often present together and can form clusters, which show a higher ability to form metastases compared to individual CTCs. Individual CTCs appear at a rate of 1 to 10 CTCs per mL of whole blood in patients with metastatic disease and have to be separated from other blood cells for efficient further studies.

These challenges need to lead to the formation of standard guidelines for the evaluation of the CTC-based diagnosis methods, which include: the efficiency of CTC isolation and detection, sample purity, and the possibility of enrichment [75]. Initiatives have been launched in Europe and America to standardize preanalytical requirements and compare different liquid biopsy methods in the same patient [76]. In the last decade, there has been an emergence of novel technologies for CTC isolation exploiting different strategies of analysis depending on the goal of the research and clinical application. However, currently, the only approved method for monitoring patients with metastatic breast, prostate, and colon cancer is the CellSearch^®^ technology.

CellSearch^®^ technology is the most widely used method for CTC analysis. The largest number of studies on different types of tumors were made using this method. It is approved for clinical monitoring of patients with metastatic breast, prostate, and colon cancer. It is currently the reference method for CTC analysis, and the results obtained with each new method are compared with those obtained using the CellSearch^®^ technology.

CTC heterogeneity and cell diversity are known phenomena and a big analytical challenge (Figure 3). In 35 % of patients, even with advanced metastatic disease, CTCs are not detectable, likely due to the epithelial–mesenchymal transition (EMT). During the EMT, epithelial cells change their phenotype to mesenchymal-like cells and lose the expression of characteristic epithelial markers (EpCAM, CK). These markers are the target of most immunoaffinity-based CTC isolation methods, including CellSearch^®^. The remaining CTCs are, thus, even further reduced in the titer and present a further limitation. Recent studies investigating the molecular characteristics of CTCs reveal that CTCs can be present in multiple phenotypes and subpopulations: epithelial, mesenchymal, and hybrid (epithelial–mesenchymal). The EMT is associated with aggressive subtypes of the malignant disease, resistance to therapy, and a worse disease outcome. Therefore, CTC isolation methods based on detecting epithelial markers miss the aggressive and invasive CTC population. Current knowledge has indicated the need for new CTC isolation methods that overcome the limitations of techniques based on the EpCAM molecule [74].

Despite the detection limitations of CellSearch^®^ technology, including its inability to detect cells that do not express epithelial markers, this method is still the gold standard and the only commercially available FDA-approved option for CTC analysis.

In addition, the CellSearch technology is, currently, relatively costly. This technology requires an experienced, trained user to confirm the results and the actual presence of CTCs in the blood. Therefore, smaller, cheaper, and easier-to-handle CTC isolation platforms, such as the ScreenCell^®^ system, are emerging.

MagSweeper^®^ is an immunomagnetic technology for isolating and enriching high-purity CTCs. The advantages of this CTC isolation method are the use of whole blood without preanalytical processing and the high purity of the isolated CTCs, which are alive and intact. MagSweeper is a method validated on clinical samples and was used for genomic profiling in studies on colorectal cancer, breast, and prostate cancer [42].

Isolation of the viable, intact cells is an important aspect of CTC analysis as it enables CTC proliferation in the cell culture and downstream analysis (Figure 5). Only viable cells can give rise to metastasis; thus, culturing CTCs postisolation can be used for: (1) the potential detection of the aggressive populations of the CTCs and their prognostic impact; (2) the discovery of novel therapies that target tumor metastasis; and (3) understanding the tumor metastasis formation. Some of the technologies that can isolate viable CTCs include MagSweeper, CTC-Chip, and ScreenCell^®^ [22,77].

Immunomagnetic cell separation enables the purification of CTCs by removing the contaminating leukocytes. The final purity of the isolated CTCs depends on the antibodies’ specificity to select the desired tumor cells. Contamination can result from the adsorption of cells in the device while being trapped between magnetic particles when using a large number of magnetic particles, as in the case of large sample volumes [71].

Microfluidic devices are compact and portable systems recognized as a powerful technology that will play a significant role in biomedical analysis in the future. Today, this field represents a new platform for the isolation and characterization of CTCs from peripheral blood. CTC isolation devices such as CTC-Chip, GEDI, and OncoCEE can be used for this purpose. The CTC-Chip uses a microfluidic system with immobilized anti-EpCAM antibodies. Microfluidic devices enable the precise control of the flow (speed and direction) of the liquid, which is essential in isolating cells that depend on the contact between the cellular antigen and the antibody. After the CTC-Chip, GEDI immunochemical microfluidic technology was developed. GEDI improved the channel system geometry for CTC isolation and was optimized for higher efficiency. The GEDI microchip combines the use of antibodies to a specific tumor antigen with hydrodynamic chromatography, which isolates cells based on size. OncoCEE is a microfluidic device that deviates from the classical isolation approach based on anti-EpCAM antibodies. This technology uses a large number of antibodies to isolate and enrich CTCs. It uses antibodies against specific tumor antigens (HER2, EGFR), but also mesenchymal markers. Samples processed with multiple antibodies simultaneously allow for the isolation of CTCs with higher efficiency, including the isolation of important EpCAM-negative cells, which would normally be missed [42]. Microfluidic devices have numerous advantages over conventional systems of “capturing” circulating tumor cells. Due to their size, they consume less reagents and energy, generate a small amount of waste, and enable high efficiency, specificity, and sensitivity in CTC isolation [78].

The ScreenCell^®^ size-based isolation of CTCs allows for rapid and simple isolation without antibodies from whole blood by microfiltration, as it uses a specialized filter with a defined pore size. ScreenCell^®^ is a simple, fast, and efficient method for CTC isolation from whole blood and is compatible with all modes of CTC analysis. The literature even shows a high isolation efficiency, by first using the ScreenCell^®^ system, and then the reference CellSearch ^®^ technology [47]. It is assumed that ScreenCell^®^ can isolate cells that do not express epithelial markers, i.e., the EpCAM molecule and those that have entered the epithelial–mesenchymal transition. Further, the ScreenCell^®^ system can isolate cells from a smaller volume (3–7 mL) of blood than CellSearch^®^ and in a shorter time, amounting to just a few minutes. Although ScreenCell^®^ is a simple method and suitable for daily routine work, it still requires preanalytical blood processing and has several additional drawbacks. There is a risk of losing very small CTCs with a diameter smaller than the pore diameter of the device microfilter. Additionally, a smaller volume of whole blood can also represent a lack of analytical accuracy, which is especially evident when the sample contains a low number of CTCs. Samples often require erythrocyte removal and lysis or blood dilution. In these methods, there is a high risk of clogging the system.

## 9. Conclusions

Circulating tumor cells are biomarkers for identifying therapeutic targets, monitoring therapy in real-time (predictive relevance), and assessing the risk of metastatic relapse or disease progression (prognostic significance). CellSearch^®^ immunomagnetic technology is the reference method and gold standard for CTC isolation and detection. The CellSearch® technology is applicable for tumors of epithelial origin. The primary feature of the technique is immunochemical detection using antibodies against the surface epithelial marker, which is the adhesion molecule EpCAM. CTCs that do not express epithelial markers or enter the epithelial–mesenchymal transition will not be isolated by this method. The Magsweeper immunomagnetic approach enables obtaining live CTCs of high purity without needing the preanalytical sample processing, which is a great advantage compared to other methods. Microfluidic devices ensure the isolation of CTCs from tiny volumes of samples with a high efficiency, specificity, and sensitivity, and with a low consumption of reagents compared to other methods. ScreenCell^®^ technology is a simple, compact, rapid, and relatively inexpensive method for isolating CTCs from whole blood samples. ScreenCell^®^ isolation depends on the size of the cells and not on the expression of surface markers. It is, therefore, applicable to all types of tumors. In methods based on the principle of microfiltration, which separates the cells based on size, there is a risk of clogging the system due to the presence of microaggregates and/or clots that prevent efficient isolation and further analysis. To introduce liquid biopsy analysis into clinical practice, there is a necessity to standardize the procedure and perform a cost–benefit analysis.

## Figures and Tables

**Figure 1 ijms-23-13582-f001:**
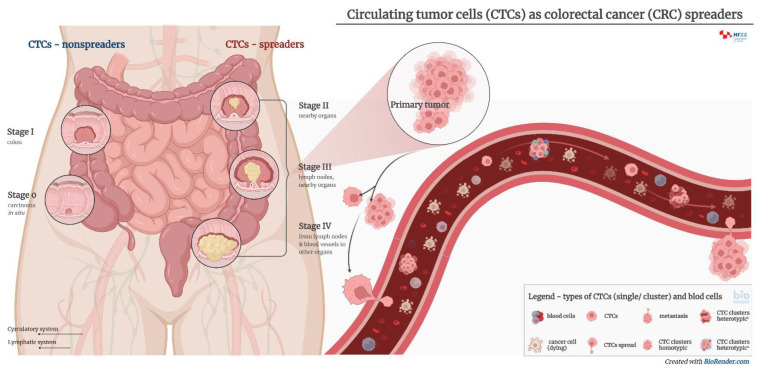
The roles of CTCs in tumor metastasis. Circulating tumor cells (CTCs) originating from primary or secondary tumors contain important information on the molecular characteristics relevant to tumor progression and therapy. CTCs can appear either as single cells or clusters, which have an increased metastatic potential and a shorter half-life. Their metastatic potential further depends on the specific microenvironment and local vasculature. CTC clusters may be further divided into two main types—homotypic and heterotypic.

**Figure 2 ijms-23-13582-f002:**
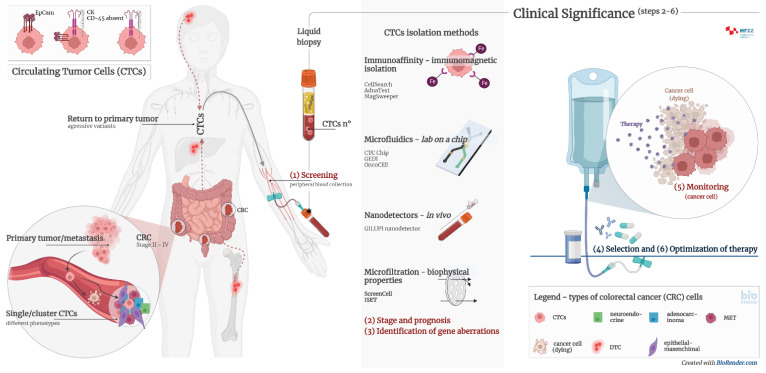
Clinical significance of CTCs and CTC clusters in diagnosis, monitoring, and optimization of therapy in cancer patients. Circulating tumour cells (CTCs) are proven to be prognostically relevant in cancer treatment in colorectal cancer (CRC). Adequate detection and characterization of CTCs can significantly contribute to cancer diagnosis, disease course prediction, therapy selection, and treatment monitoring. (1) Screening—detection of cancer patients by determining the number of CTCs and/or CTC clusters from peripheral blood; (2) stage and prognosis—detection of patients with an increased risk for the development of metastases by determining the number of CTCs and/or CTC clusters from peripheral blood; (3) identification of gene aberrations in CTCs important for therapy, such as mutations in KRAS, APC, EFGR, etc.; (4) selection of therapy—CTC cultivation and selection of effective pharmacotherapy; (5) monitoring—determining the number of CTCs and genome analysis to detect transcriptional changes; (6) optimization of therapy—monitoring the effectiveness of pharmacotherapy and early detection of disease remission by determining the number of CTCs and genome analysis.

**Figure 3 ijms-23-13582-f003:**
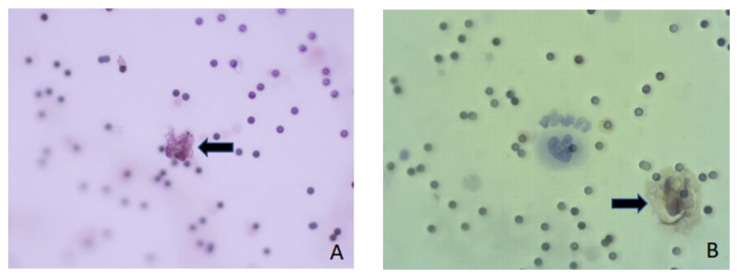
Cells were analyzed based on morphological characteristics or immunological features (magnification, ×400) (**A**); isolated cells stained with anti-pan-CK antibodies (brown; magnification, ×400) (**B**).

**Figure 4 ijms-23-13582-f004:**
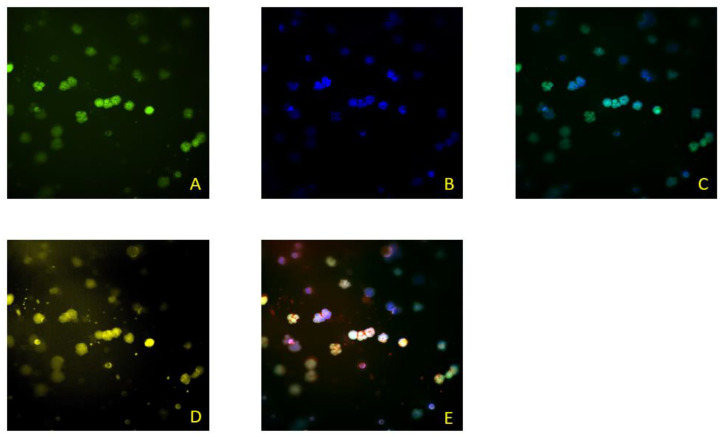
ScreenCell^®^—CTCs were IF stained with anti-pan-CK antibodies (green, (**A**)); DAPI staining was performed for nuclei staining (blue, (**B**)); composite of (**A**,**B**) micrographics (**C**); leucocytes IF stained with anti-CD45 antibodies (yellow/red, (**D**)); and a composite of (**A**,**B**,**D**) micrographics (**E**).

**Figure 5 ijms-23-13582-f005:**
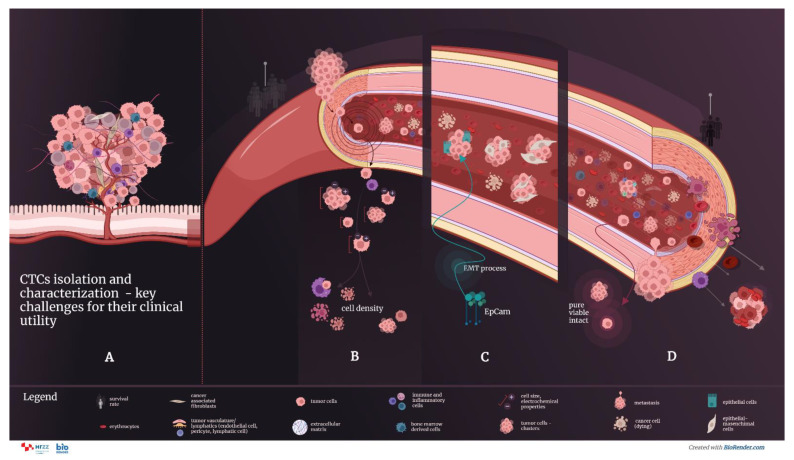
CTC isolation and characterization are key challenges for their clinical utility. (**A**) The complexity of the primary tumor microenvironment affects tumor progression and CTCs and CTC cluster formation. CTCs detach from the primary tumor as single cells and clusters, shed into the bloodstream, and migrate to colonize in distant organs, known as metastasis. The predictive significance of CTC analysis includes the detection of CTCs in the early stage of malignancy in resectable solid tumors.The CTC isolation methods are based on the differences between target CTCs and blood cells. These differences include physical properties (size, density, electrical charge, and deformability) and specific biological characteristics (expression of surface protein markers and viability).The specific features of CTCs make their isolation methods very demanding: (**B**) The low concentration of CTCs in the blood (estimated as 1–10 single CTCs and 1 CTC cluster per 1 mL of blood) and their extremely short life span in circulation (a few hours for CTCs and even shorter for CTC clusters); (**C**) CTC heterogeneity and cellular plasticity due to the epithelial–mesenchymal transition (EMT), in which the expression of characteristic epithelial markers, the target of most immunoaffinity-based CTC isolation methods, is lost; (**D**) in the circulation system, CTCs are subject to blood shear force damage and immune attack. Isolation of the viable, pure, and intact cells is an important aspect of CTC analysis as it enables CTC proliferation in cell culture and downstream analysis.

**Table 1 ijms-23-13582-t001:** Clinical significance of CTCs in colorectal cancers.

Number of Patients	Detection Method	CTC No. (%)	Clinical Significance	Ref.
34	Multiplex PCR	20 (59)	Therapy alignment and monitoring; CTCs could predict chemotherapy response; moreover, EGFR status of CTCs could predict the likelihood of targeted therapy response.	[48]
30	Density gradient centrifugation, CK20 qRT-PCR and immunomagnetic CTC number determination	30 (100)	CTC number reflects the chemotherapeutic sensitivity of CRC patients. Microscopic CTC single-cell, doublet, and cluster numbers were found in correlation with CK20 qRT-PCR results.	[49]
40	CELLection Dynabeads^®^	27 (68)	Therapy alignment and monitoring. Significant shorter progression-free survival (PFS) was found in patients with CTCs positive for the expression of ALDH1, survivin and MRP5.	[50]
467	CellSearch	467 (100)	Therapy alignment and monitoring; CTC count provides additional information to CT imaging for early recurrence monitoring.	[51]
141	RT-PCR	141 (100)	Therapy alignment and monitoring; CTC persistence after surgical resection was a significant marker for early recurrence.	[52]
14	CellSearch	14; 4 (29) after chemotherapy	Therapy alignment and monitoring; CTC-negative patients after chemotherapy had significantly better treatment response.	[53]
42	CellSearch	22 (52.3)	Patients with CTCs ≥3/7.5 mL may benefit from the intensive 4-drug regimen (irinotecan, oxaliplatin, and tegafur-uracil with leucovorin and cetuximab).	[54]
61	CellSearch	27 (44.3)	CTC heterozygosity and heterogeneity exist in KRAS status among CTCs within all patients and between CTCs and tumor tissues.	[55]
66	CanPatrol Multiplex mRNA-ISH	57 (86.4)	CTC count ≥6/5 mL was associated with decreased PFS and OS. LGR5 expression in CTCs may serve as a marker for CRC metastasis.	[56]
138	ISET device-CTCBIOPSY	63 (45.7)	Postcurative resection CTC count > 1/2.5 mL was associated with shorter 3-year RFS rate.	[57]
91	CanPatrol mRNA-ISH	51 CTC (56.0); 46 mCTC (50.5)	Mesenchymal CTC count ≥1/5 mL and COX-2 expression in mCTCs were associated with distance metastasis.	[58]
34	Microfluidic chips	34 (100)	Therapy alignment and monitoring; comparison of mutational status of CTCs, ctDNA, and primary tumor tissue revealed great heterogeneity.	[59]
130	MACS	67 (51.54)	Postoperative CTC count ≥2/3.2 mL in non-mCRC was associated with decreased RFS.	[60]
106	MACS	100 (94)	HAI/target therapy with drugs selected by liquid biopsy precision oncotherapy is a safe and efficacious alternative therapeutic strategy for unresectable colorectal liver metastases patients.	[61]
21	ScreenCell^®^	21 (100)	Isolation of CTCs by size (as a label-free technique with subsequent immunofluorescence labeling) gives a very high detection rate.	[62]
21	CK20 RT-qPCR	15 (71.4)	The CK20 RT-qPCR method gives a relatively high detection rate.	[62]
21	NYONE^®^	11 (52.4)	Application of a semiautomated microscopic approach with NYONE^®^, an examiner-independent procedure for CTC detection.	[62]
50	CellSearch	46 (92)	CTC counts ≥3/7.5 mL at baseline and day 21 after initiation of regorafenib were associated with decreased PFS and OS. Patients had significantly increased EGFR expression at day 21 and/or PD compared to baseline.	[63]
589	CellSearch	241 (41)	Baseline CTC counts ≥3/7.5 mL were associated with clinical or pathologic features associated with poor prognosis.	[64]
7	ScreenCell^®^, Immunofluorescence Staining	7 (100)	Promising test for the future isolation and characterization of different CTC subtypes, including clusters.	[65]

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
