# Peer review of "Circulating Tumor Cells in Colorectal Cancer: Detection Systems and Clinical Utility"

_ijms, 2022, doi:10.3390/ijms232113582_

Round 1

Reviewer 1 Report

The review (completed with some own pilot experiments) focuses on a hot topic in cancer diagnostics and follow up. The literature part of the manuscript adequately describes the significance and problematics of fluid biopsy techniques on a wide scale (various analytical tools). The authors discuss the advantages and disadvantages of genetical methods and cellular approaches for isolation and characterization of tumor related markers with a main focus on CTCs. The review is based on a large scale of understanding the significance of CTC evaluation including dormant tumor cells (DTCs). In my opinion, the text should be completed with the recent problematics of the existence of tumor stem cells (TSCs) because these cells show very high genome instability and resistance to chemotherapy with a potency to hide in the body in dormant phase for a long time (e.g. in breast cancer). The TSCs may also be identified if captured by using stem cell markers. Also, the authors should discuss which techniques enable the examination of the viability of CTCs because only live cells can give metastases. It would be nice to read about the possibility (if there is any) of culturing the captured CTCs which hypothetically makes the study  of their drug resistency and metastatic potential possible. 

The Discussion section I would put at the end of the text before the Conclusions.

The authors show some results obtained in their laboratory using a cell exclusion filter technique. However, the number of the studied patients and their demographical/stage data are not indicated in the text. The figures are not too informative, I suggest to add arrows to the appropriate stainings. In figure 4. the CTC number seems to be too high when stained for CK. Furthermore, the anti pan-CK is not specific to tumor cells only. The patient's data is also missing from the legend.

The English of the manuscript requires a thorough check and correction because some sentences are very unclear.

Author Response

Thank you for your useful comments on our article.

In accordance with the reviewer's requests, the text was supplemented with recent problems related to the tumor stem cells and with techniques that enable monitoring the viability of CTCs. Cultivating CTC is a very interesting field, but for now, unfortunately, we do not have our own results on this topic. The text is edited in such a way that the Discussion is placed at the very end before the Conclusions. Illustrative views have been added to the manuscript. Paragraph a has been adapted to its own results,  and arrows in appropriate colors have been added. The entire manuscript was thoroughly checked and unclear sentences were corrected.

Reviewer 2 Report

Very intresting work. I really did enjoy by reading this manuscript.

Author Response

The English language and style were corrected in the manuscript. Special thanks to you for your good comments.

Reviewer 3 Report

In this review, Petrik et al give an update on the current state and perspective on the role of circulating tumor cells in colorectal cancer. While this topic of the review is not particularly unique, the authors give a solid overview on the topic and the article could be used as a reference for other researchers in the field.

In addition to the actual review, the authors also show some own data on CTC enrichment in colorectal cancer patients using the Screen Cell device (paragraph 8). Honestly, the intention of this paragraph is not really clear to me, as in principle only the workflow with some original figures are presented but without any significant data or analysis. Personally, I would just delete this paragraph as it does not really contribute to the review article. Or, alternatively, add some substantial data, e.g. comparing the results from the study (fo example CTC quantifcation, positivity rate) with other pubished data using the same or different technologies.

Additionally, if the study should be included, I would ask not only for a statement that patients consented to th analysis, but additionally ID of the ethics vote and location of the responsible ethics committee.

Before publishing, the article would also need some language editing, I I would suggest to give the manuscript to a native speaker for thorough check of language and grammer, as well as smoothening of some sentences.

Author Response

Thank you for your useful comments on our article.

Obliging to the reviewer's requests, we thoroughly checked the English language of the manuscript. We have also adapted paragraph 8 "Assessment of ScreenCell Cyto kit" to be more relevant to the topic of the manuscript itself.

Round 2

Reviewer 1 Report

The authors impoved the manuscript a lot. There is a ceraful review on CTCs related to epithelial cancers with available methods for isolation of CTCs from bood. The authors evaluate the potential clinical application of CTC number and also give a focus on other tumor-related markers (cell-free dNA and others). The review is now very carefully written and the experimental part is placed as an example for CTC technologies and is not given as a fully evaluable clinical example.